# Accretion Disks and Long Cycles in $\beta$ Lyrae-Type Binaries

R. E. Mennickent 

Departamento de Astronomía, Universidad de Concepción, Concepción 4030000, Chile; rmennick@udec.cl

**Abstract:** In order to inquire about the nature of the accretion disks formed around the more massive companion in binaries with $\beta$ Lyrae-type light curves, we review literature presenting some physical and observational properties of these systems. In addition, we inspect the photometric time series of three representative eclipsing systems obtained by the Optical Gravitational Lensing Experiment (OGLE) project during the last decades and compare them with $\beta$ Lyrae. All these three systems show indications of being semidetached with a more massive B-type component and in a mass transfer stage. They also show long photometric cycles, and two of them show changes in the orbital light curve that can be interpreted in terms of structural changes of the accretion disks, eventually driven by variations in the mass transfer rate.

**Keywords:** binary stars; circumstellar matter; mass transfer; accretion disks

## 1. Introduction

The system $\beta$ Lyrae is an interesting testbed for models of accretion flows by Roche-lobe overflow occurring in intermediate mass semidetached binaries. This binary consists of a B8 II "donor" star transferring mass by a gas stream onto a more massive "gainer" star of probably also B-type. The gainer is not big enough to fully intercept the stream that revolves around it to finally form an optically thick and geometrically thick disk [1]. In addition, a jet has been detected emerging from the region of interaction between the disk and gas stream [2,3]. The first detection of interferometric photospheric tidal distortion due to the Roche lobe filling was reported by Zhao et al. [4] in $\beta$ Lyrae, based on interferometry performed with the Center for High Angular Resolution Astronomy (CHARA) array. These authors also detected the thick disk surrounding the gainer in $H$-band images. The level of activity in $\beta$ Lyrae is larger than in ordinary Algols, as revealed by variable eclipse shapes, super-orbital photometric cycles, and orbital period changes. The orbital period of 12.9 days increases at a rate of 19 s/yr [5] and this is usually interpreted as caused by the transfer of mass from the donor at a rate of $2.2 \times 10^{-5}$ M$\odot$/yr [5,6]. The emission in H$\alpha$ and other low ionization ions has been modeled in terms of an extended atmosphere of the disk, two perpendicular jets expanding at $\sim$700 km s$^{-1}$, and a symmetric shell with the radius of $\sim$70 R$\odot$ [7]. The light-curve model is significantly better with two bright regions in the disk rim with temperatures 10 and 20 per cent higher than the disk outer edge temperature [8]. $\beta$ Lyrae is the prototype of eclipsing binaries classified EB according to the shape of their light curves; unequal minima and rounded inter-eclipses shoulders, a signature of a tidally distorted donor star due to the gravitational interaction of the gainer. $\beta$ Lyrae is an Algol system from the evolutionary point of view, i.e., a system that has reversed its mass ratio as a result of the mass transfer from the initially more massive star onto the initially less massive star. For the above reasons, some critics have mentioned using $\beta$ Lyrae as a prototype of a single class of binaries [9]. However in this article we follow the traditional approach, grouping together the systems with similar light curves *that also show* long cycles.

The existence of long cycles seems to be usual in $\beta$ Lyrae-type binaries; Gaposchkin (1944) reported a long cycle of 517.6 days in RX Cas, Lorenzi [10] showed the presence of a 411 day long cycle in AU Mon, Hill et al. [11] reported a long cycle of 322.24 days for V360,

and Lac and Koubsky et al. [12] reported a long cycle of few hundred days in CX Dra. More recently, a period of 253.4 days was reported for V 393 Sco [13] and that of ∼270 d was found in UU Cas [14]. Guinan [15] inferred a 275-day period for $\beta$ Lyr attributing a pulsational origin to the B8 II star, although he also mentioned possible changes in the structure of the disk. Studying $\beta$ Lyrae, Harmanec et al. [2] considered the 282-day cycle as a possible beat between the orbital period and the 4.7-day period detected in spectroscopy. They also noticed the similarity of $\beta$ Lyrae with V 1343 Aquilae (SS 433), a massive X-ray binary with an orbital period of 13.08 days and bipolar jets. Posteriorly, Wilson and van Hamme [16] conclude that neither apsidal advance nor precession can account for the 282-day light variation in $\beta$ Lyrae, but they were unable to exclude disk pulsations as the origin.

From a theoretical point of view, the mass transfer stage of binaries with at least one B-type component usually allows the transfer of few solar masses onto the gainer, modifying its chemical composition and rotational velocity, eventually forming a relatively massive disk around it [17]. As an example, for $\beta$ Lyrae, a disk with about 4% of its central star's mass was estimated [18]. The gainer is expected to be accelerated rapidly toward critical rotation, which should favor the stop of accretion onto the stellar surface and the accumulation of mass in the disk, although after the mass transfer episode tidal forces and magnetic braking should slow down the rotational velocity [19–21]. The process should allow the formation of an accretion-decretion disk, a configuration that recently has been explored from the theoretical side [18]. How much mass is lost into the interstellar medium by this process of mass transfer is currently unknown [22], but we should expect that the disk responds to different mass transfer rates when the interaction episode evolves, finally changing its physical structure.

Recently, hundreds of $\beta$ Lyrae-type binaries (i.e., those with EB-type light curves) have been found showing long photometric cycles in extensive surveys of variable stars performed in the Galaxy, the Galactic Bulge, and the Magellanic Clouds; they are often named Double Periodic Variables (DPVs) [23–26]. After summarizing the physical and observational properties of the disks found in these systems, we review the evidence from recent research suggesting that in some cases the long cycles could be used as diagnostics of changes of a disk physical structure and variability. Contrary to $\beta$ Lyrae, the orbital period of most of these systems is relatively constant, suggesting a rather mild mass transfer rate or a compensating loss of angular momentum through winds or equatorial outflows [27,28].

## 2. The $P_o$-$P_l$ Relationship

We reviewed the literature and the result of the compilation of orbital and long periods for 218 $\beta$ Lyrae-type binaries as shown in Figure 1 and 2. Some properties of the DPVs derived in previous years by different authors have been already summarized and we give here just a brief overview [29] and references therein. They usually consist of a B-type gainer and later type giant transferring mass through the inner Lagrangian point by the Roche lobe overflow. Most studied systems show the size of the gainer compatible with a tangential or near tangential impact of the accretion stream. The disks are luminous enough to contribute significantly to the continuum light curve, and are mostly optically thick and sometimes geometrically thick. The disks are not energized by accretion as in Cataclysmic Variables, but by the illumination of the gainer [30,31]. A cooler and optically thin chromosphere around the optically thick disk is inferred from the presence of Balmer emission lines in several systems. Chromospheric lines are sometimes observed in the donor of some binaries, suggesting an origin in a magnetic field; the line emission is often stronger during the high state of the long cycle. The long cycle has a larger amplitude in redder bandpasses and sometimes shows a single hump and sometimes a double hump. In some systems, the long cycle changes its length and amplitude. DPVs are more luminous, hotter, and more massive than classical Algols.

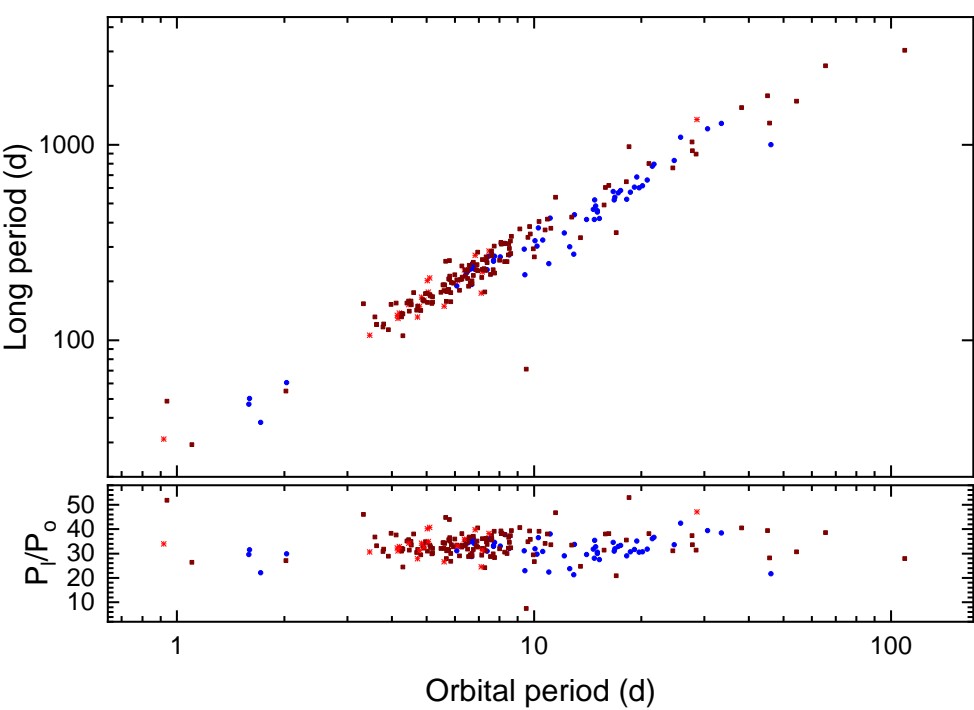

**Figure 1.** Orbital periods and long cycle lengths for systems similar to β Lyrae in the Galaxy (blue dots), Small Magellanic Cloud (red asterisks), and the Large Magellanic Cloud (brown squares). Data from [23–26,32–34].

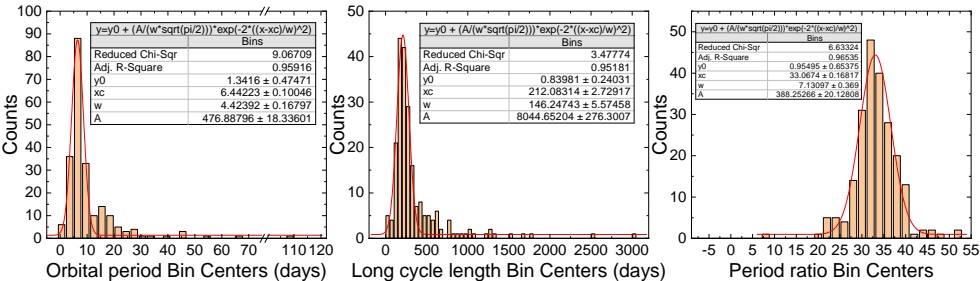

**Figure 2.** Period and period ratio distributions and best Gaussian fits for the data of Figure 1.

A dynamo operating in the donor has been proposed as the cause for the long cycle [35]. In this model, based on the Applegate mechanism [36], the dynamo changes the quadrupole moment and therefore the shape of the donor producing at certain epochs larger expansion through the inner Lagrangian point hence increasing the mass transfer rate; this extra supply of material is injected into the disk, eventually increasing the temperature and luminosity of the stream-disk impact region, and hence increasing the hotspot wind. This procedure should also affect the disk thickness, producing variable occultation of the gainer impacting the total system brightness. For the seven binaries studied with orbital periods between five and 13 days, the average deviation between predicted and observed period ratio is about 12%. For the three binaries with longer orbital periods, the deviation ranges from 26% to about 55% [35]. Some additional support for a magnetic origin of the long cycle comes from finding that the dynamo number, a measure of magnetic activity driven by convective and rotational motions inside the star, increases during epochs of mass transfer [37]. Spectropolarimetric studies of these objects are practically nonexistent, but scarce investigations oriented to test the circumstellar matter suggest a promising role for this technique in future investigations [38].

Another compelling explanation for the long period is disk precession, since it causes periodic flux variations as the disk area projected in the direction of the observer line of sight changes. Disk precession has been discussed in terms of its influence on accretion [39]

and models of disk warping and precession were presented [40]. These models give the predicted precession period for a disk with a constant surface density and given disk radius as a function of the orbital period and mass ratio of the binary. The predicted period ratio for typical DPV mass ratios and for large disks is a few tens, in principle compatible with the observations. However, although disk precession was presented in the beginning as a possible explanation for the long cycle [23], some later observations put doubts about its feasibility. For instance, DPVs with disks whose radius is well below the tidal radius are observed (Figure 3), and it is difficult to understand how these disks could experience precession and warping by the tidal influence of the donor star. In addition, large amplitude long cycles are observed in some eclipsing systems, where the projected disk area is in principle small. Finally, in the case of V 393 Sco, the constancy of the asymmetry observed in the double peak H$\alpha$ emission during the long cycle, as well as the behavior of its peak separation, and the constancy of the shape of the orbital light curve, have been mentioned as arguments against disk precession [41].

### 3. Insights on Disks of Systems Showing $\beta$ Lyrae-Type Light Curves

The stream of mass from the donor star revolves around the primary. If the stream is small enough, it forms an accretion ring that spreads because of viscosity to rapidly form an accretion disk [42]. The radius where this phenomenon occurs is dubbed the "circularization radius". This radius, the external disk radius, and the radius of the gainer can be normalized to the binary orbital separation and visualized as a function of the mass ratio alone. It is clear that the $\beta$ Lyrae-type systems with long cycles are found in the zone of tangential impact, where the stream hits the gainer quasi-tangentially, eventually transferring a large amount of angular momentum to it and spinning it up (Figure 3). Some disks extend up to the tidal radius, where tidal forces disrupt its structure. Many measurements of the disk radii were derived from light curve models susceptible to the optically thick parts of the disk, so it is reasonable to expect discrepancies with methods sensitive to the optically thin portions of the disk; i.e., methods using the emission line widths or peak separation, for instance. A good example of this mismatch is the radius of the disk of AU Mon. While Atwood-Stone et al. [43] determine 23 $R_\odot$ with a technique sensitive to optically thin line emitting regions, Djurašević et al. [44] found 13 $R_\odot$ with a method sensible to flux emitted in the continuum, i.e., in inner, denser, and hotter regions of the disk.

Another interesting finding about the disks is related to their thickness. Models indicate that the disks are thicker than expected from a regime of hydrodynamical equilibrium. This has provoked the inference of the existence of vertical turbulent motions capable of rapidly transporting the thermal energy and increase in the disk vertical height [7,45].

In this context, it is valid to wonder about the nature of hot and bright zones detected in several systems, through the technique of Doppler tomography, eclipse mapping, or intensity maps derived from light curve models [30,46,47]. One of these zones is located near the theoretical region of stream-disk impact and it is relatively stable in position, the other region is found on the opposite side of the disk and shows larger variability in position [29,45]. This last region has not been fully explained, although it could be associated to vertical oscillations of the gas at the outer edge of the accretion disk due to interaction with the stream of matter [48]. The persistence of these hot zones suggests the continuous transfer and release of kinetic energy from the donor through the gas stream during the process of mass exchange.

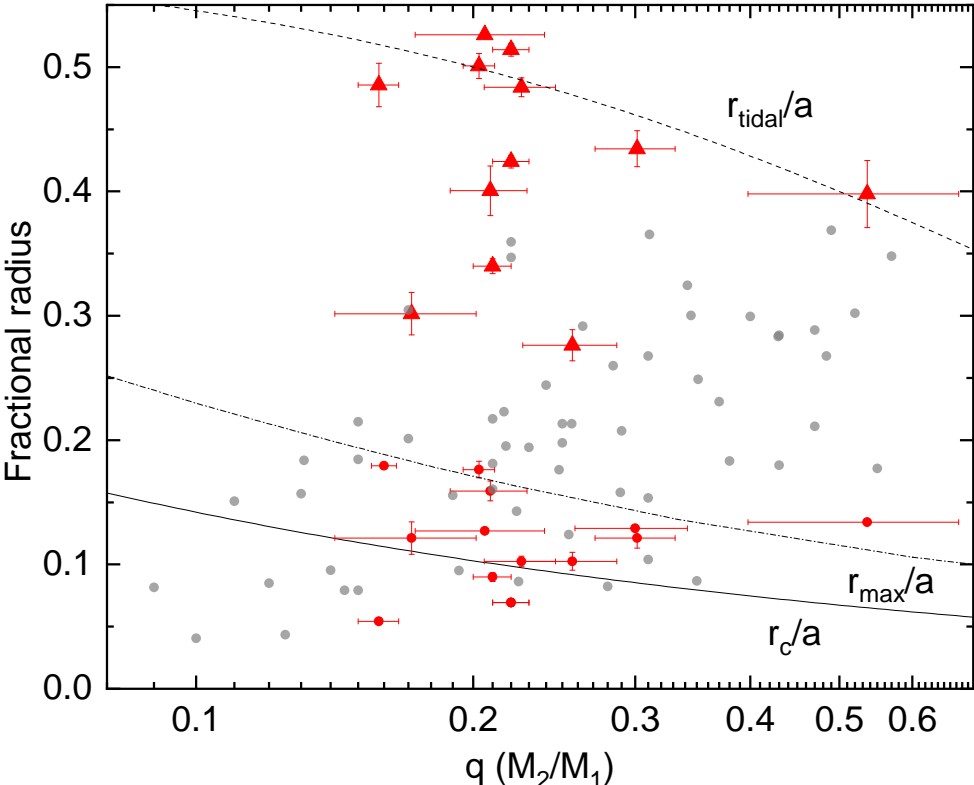

**Figure 3.** The fractional radius for the gainer (R1/a; circles) and disk (Rd/a, triangles) of $\beta$ Lyrae-type binaries, with *a* as the orbital separation. Below the circularization radius shown by the solid black line, a disk should be formed and below the dash-point a disk might be formed. The tidal radius indicates the maximum possible disk extension (upper dashed line). Semi-detached Algol primaries from [49] are also shown as gray points as comparison. Adapted with permission from [24].

Possibly related to the disk hot regions are the jets or winds observed in some systems. Apart from the jet reported in $\beta$ Lyrae mentioned before, a bipolar wind was reported in V 393 Scorpii [41]. This wind has a larger line and continuum emissivity at the high state. The system shows highly variable chromospheric emission in the donor, as revealed by the Doppler maps of the emission lines Mg II 4481 and C I 6588. Notable and novel spectroscopic features like discrete absorption components, especially visible at blue depressed O I 7773 absorption wings during the second half-cycle are observed [41]. Winds were reported in HD 170582 and HD 50526 based on persistence of the blue emission components in the H$\alpha$ line [50,51]. Doppler tomography of the H$\alpha$ line shows larger hotspot emission during the brighter state of the long cycle in HD 170582 [30]. In the past, winds were inferred from the study of ultraviolet lines of Si IV, C IV, O VI, and N V in Algol-related systems [52,53]. According to recent models, some of the jets and winds might be anchored to the hotspot and driven by the spot's luminosity and others linked to the hot star photosphere [17,54].

## 4. Light Curve Models for Individual Systems

In this section, we give an overview of the main observational features of three interesting eclipsing binaries of the DPV type, OGLE-LMC-DPV-097, OGLE-BLG-ECL-157529, and OGLE-LMC-DPV-065, and compare them with $\beta$ Lyrae. Some physical parameters of these systems are given in Table 1. The data sampling from the Optical Gravitational Lensing Experiment (OGLE) project provides an homogenous and high-quality dataset useful for exploring temporal variability on time scales of days, weeks, months, and years [55,56].

**Table 1.** Systems summarized in this article ordered by decreasing orbital period. Those showing large changes in the shape of the orbital light curve are indicated with asterisks. Indexes 1 and 2 refer to the gainer and donor, respectively.

| System | $P_O$ (d) | $P_l$ (d) | $M_1$ (Msun) | $M_2$ (Msun) | $T_1$ (K) | $T_2$ (K) | i (°) |
|---|---|---|---|---|---|---|---|
| *OGLE-BLG-ECL-157529 | 24.80 | 900–800 | 4.83 | 1.06 | 14,000 | 4400 | 85.5 |
| $\beta$ Lyrae | 12.95 | 282.4 | 13.2 | 2.97 | 30,000 | 13,300 | 86.1 |
| OGLE-LMC-DPV-065 | 10.03 | 350–210 | 13.8 | 2.81 | 25,460 | 9825 | 86.7 |
| *OGLE-LMC-DPV-097 | 7.75 | 306 | 5.51 | 1.10 | 14,000 | 4030–6870 | 74 |

The above systems have been recently studied and revealed interesting clues on changes of accretion disk properties during the long photometric cycles [45,57,58]. For the model of the orbital light curve, a fitting was performed using the inverse-problem solving method based on the simplex algorithm. The model of a binary system with a disk was used to find the best solution for the stellar and disk parameters, assuming some fixed parameters, usually the mass ratio and the donor temperature derived from spectroscopic data or determining the mass ratio $q$ using a photometric method in the absence of spectroscopic data [59]. The model has the strength of following the changes of disk properties like temperature, inner and outer thickness and radius, along with the location, size, and temperature of hot and bright regions. However, it has the weakness of neither explicitly including stellar or disk winds, nor equatorial mass outflows [60,61].

OGLE-LMC-DPV-097 is characterized by a long cycle length of ∼306 d and an amplitude of ∼0.8 mag in the *I*-band. The system has an orbital period of 7.75 d and relatively small stellar masses, 5.5 and 1.1 M$_\odot$. The disentangling between the long cycle and orbital cycle reveals the most remarkable lesson from the study of OGLE-LMC-DPV-097, namely that the orbital light curve changes in a systematic way during the long cycle. At the minimum of the long cycle, the secondary eclipse practically disappears and during the ascending branch the system is brighter in the first quadrant than during the second quadrant. This can be understood in terms of changes in the temperature and size of the disk along with changes in the hot and bright spots. The disk radius is 7.5 $R_\odot$ at the minimum and 15.3 $R_\odot$ at the ascending branch; its temperature at the outer edge changes from 6870 to 4030 K in these two stages (Figure 4) [57].

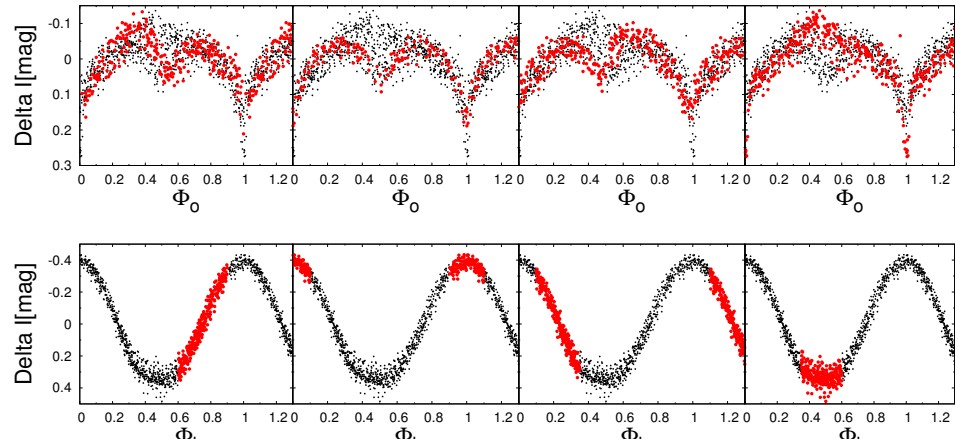

**Figure 4.** OGLE-LMC-DPV-097. Disentangled long-cycle (**down**) and orbital (**up**) light curves phased with the respective periods. Black dots show the complete dataset, red dots show segments of the data of the long-cycle. Figure 1 from [57] reproduced with permission.

OGLE-LMC-DPV-065 is another interesting DPV with a large amplitude of the long cycle. It has stellar masses similar to those of $\beta$ Lyrae and the 10.03-d orbital period

compared to 12.95 d in the latter system. The long cycle is characterized by a double hump light curve in the *I* and *V* bands with amplitudes of ∼0.3 and ∼0.2 mag, respectively, whose general shape is nearly constant with only minor variations. After a continuous decrease of the long-period from 350 to 218 d during about 13 yrs, the long cycle remained almost constant for about 10 yrs. However, the orbital light curve is pretty constant in shape and no clear link between the disk structure and long cycle has been observed for this system, in a clear contrast to OGLE-LMC-DPV-097. Therefore the object places a challenge for the interpretation of the long cycle just in terms of structural changes of the disk geometry (Figure 5) [58].

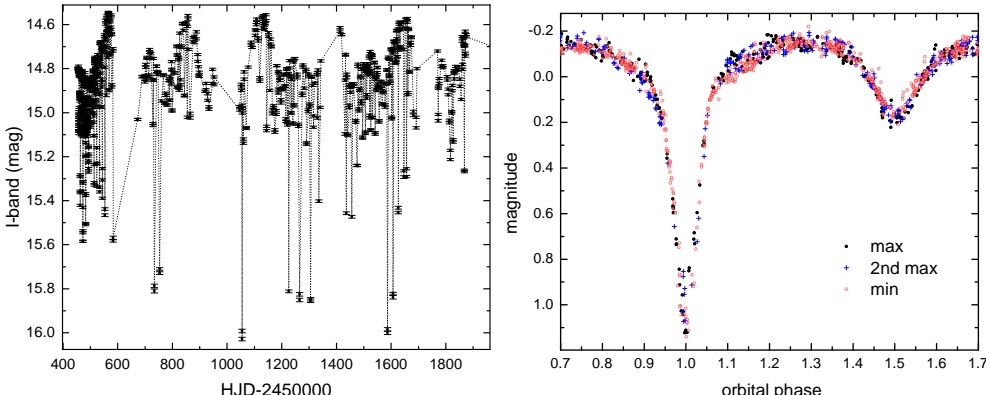

**Figure 5.** OGLE-LMC-DPV-065 shows a double hump long cycle with primary and secondary maxima and minima. **Left**: A zoom in the light curve showing the long cycle and orbital variability. **Right**: Disentangled orbital light curve showing eclipses during main and secondary long cycle maxima ($0.9 < \Phi_l \leq 1.1$ and $0.4 < \Phi_l \leq 0.6$, respectively) and minima ( $0.2 < \Phi_l \leq 0.4$ and $0.6 < \Phi_l \leq 0.8$). Vertical axis shows the differential *I*-band magnitude. Adapted from Figures 1 and 4 with permission from [58].

When we look at the photometric behavior of OGLE-BLG-ECL-157529, the situation again turns out to be very interesting. This system is characterized by orbital and long periods of 24.8 d and ∼850 d and the stellar masses are comparable to those of OGLE-LMC-DPV-097. We observe that the magnitude of the primary minimum is practically constant whereas the secondary minimum shows large changes. Let us remember that the secondary minimum is due to the occultation of the donor by the structure formed by the gainer and disk. This finding immediately points to a variable accretion disk. One interesting finding is that the magnitude at orbital phase 0.25 follows and traces the long cycle pretty well. The disk fractional radius ($F_d$, a measure of the filling of the gainer Roche lobe) indicates that the disk is larger than the tidal radius at maximum activity and longer cycle. However, $F_d$ becomes smaller than the tidal radius when the long cycle is faster and its amplitude is smaller on later epochs (Figure 6). In this system, the minimum of the long cycle can be explained as a thicker disk, occulting a larger fraction of the gainer and a larger mass transfer rate implies a hotter disk in the outer edge (Figure 7). The oscillations of the disk radius on timescales of hundreds of days cannot be explained by the release of viscous energy as occurs with superhumps in precessing disks in cataclysmic variable stars of the SU UMa type. This is because Lindblad resonance regions are far beyond the disk radius. The alternative of injections of variable amounts of mass as cause for long photometric variability has been proposed [45].

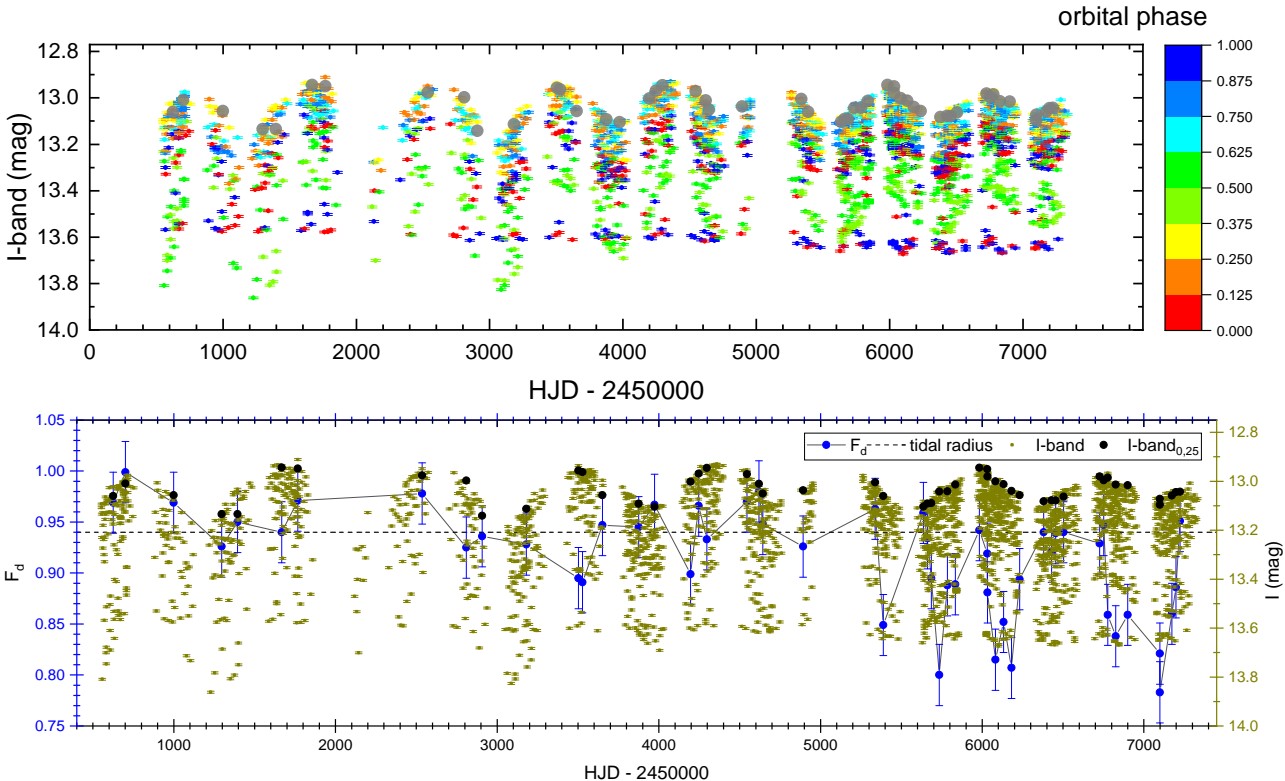

**Figure 6.** **Up**: The OGLE *I*-band light curve of OGLE-BLG-ECL-157529. Colors indicate ranges of orbital phases. Gray dots show the magnitude at orbital phase 0.25 for 52 consecutive data subsamples. **Down**: Fractional disk radius, *I*-band magnitude, and *I*-band at orbital phase 0.25. The tidal radius is indicated by a horizontal dashed line. Figure 1 and 10 with permission from [45].

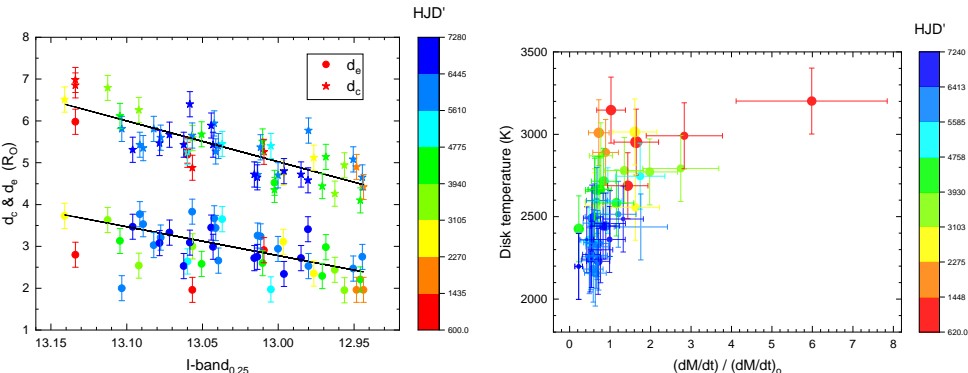

**Figure 7.** **Left**: Dependence of disk vertical thickness with the system magnitude of OGLE-BLG-ECL-157529 at $\Phi_o$= 0.25. Best linear fits are also shown. $d_c$ and $d_e$ are the disk vertical thickness at the inner and outer radial border, respectively. **Right**: Disk temperature versus mass transfer rate for the same system. HJD' = HJD-2 400 000. Figures 5 and 7 with permission from [45].

## 5. Conclusions

Disks found around the more massive stars of binaries showing light curves similar to *β* Lyrae show signatures of activity due to the mass transfer process. It is not clear how massive these disks are, but they are optically thick and usually geometrically thick, sometimes hiding the central star and perturbing the radial velocities of lines normally associated to this star. The disks vary in size, thickness, and temperature and show bright zones, and in a few cases, jets and outflows have been reported. The disks are fed by a Roche lobe overflow from the donor star, and the disk changes can in principle be explained by variations in the mass transfer rate. Zones of low density, where Balmer emission lines

form, have also been reported and revealed in Doppler maps. These disks are heated by the radiation of the gainer star and possibly by the injection of energy by the impact of the gas stream. Accretion plays a minor role as a source of disk luminosity because of the shallower gravitational potential around the gainer, compared with compact objects, whose disks luminosity is accretion driven. It is not clear what happens in the interaction zone between the inner disk and gainer photosphere. Due to the blending of gainer lines by disk emission/absorption, it is hard to measure the true rotational velocity of the gainer—and hence probe critical rotation—measuring the spectral line broadening in these systems.

Interestingly, some of the binaries classified as EB or a $\beta$ Lyrae-type in terms of its light curve, also show a long cycle as $\beta$ Lyrae does, and show changes in the shape of the orbital light curve that can be explained as structural modification of the accretion disk. However, unlike $\beta$ Lyrae, most of these binaries show a constant orbital period that indicates that $\beta$ Lyrae is a rather unusual object in the class. This could be explained if $\beta$ Lyrae is found in a rare and unusually high mass transfer state while the other systems are found in milder and more common stages of mass transfer. Therefore $\beta$ Lyrae itself is far from being an appropriate prototype of $\beta$ Lyrae-type systems. Another finding is that, whereas disk thickness changes can explain the long cycle in terms of differential occultation of the gainer in some systems, as changes in disk size and temperature as well, this seems not to be applicable to all systems. A magnetic dynamo that periodically changes the equatorial radius of the donor star has been proposed as a method for cyclic mass transfer and as an explanation for the long cycle. However, the magnetic nature of the donor stars in DPVs has not been established. On the other hand, the alternative hypothesis of precessing disks faces observational challengers in some systems, hence the enigma of the long cycle remains unsolved and deserves further investigation.

**Funding:** This research was funded by ANID PIA/BASAL FB210003 and FONDECYT 1190621.

**Institutional Review Board Statement:** Not applicable.

**Informed Consent Statement:** Not applicable.

**Acknowledgments:** The author thanks the referees who contributed to improving the first submitted version of this article. The author acknowledges R.E. Wilson, Gojko Djurăsević, and Juan Garcés for their valuable comments on an earlier version of this manuscript. The author also thanks the support of ANID PIA/BASAL FB210003 and FONDECYT 1190621.

**Conflicts of Interest:** The author declares no conflict of interest.

## Abbreviations

The following abbreviations are used in this manuscript:

DPV     Double Periodic Variable
OGLE    Optical Gravitational Lensing Experiment

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
