# Peer review of "Accretion Disks and Long Cycles in β Lyrae-Type Binaries"

_galaxies, doi:10.3390/galaxies10010015_

Round 1
Reviewer 1 Report
My comments and suggestions are described in the attached document.

Author Response
We thank the referee for the time invested in reviewing this manuscript. I have made all the correction requested by the referee and they are marked in boldface in the new version. Here is the list of changes made:
Abstract.
Lines 5-6
"All these three systems show indications of being semidetached with a B-type more massive component and in a mass transfer stage."
Correction:
All these three systems show indications of being semidetached with a more massive B-type component and in a mass transfer stage.
R: done
Line 7
"... and two of them shows changes ..."
Correction: "... while two of them show changes ..."
R: done
Line 8
"In these two cases the observations can be observed as ..."
Correction: "In these two cases the observations can be interpreted as ..."
Comment to the last two sentences of the Abstract: These statements look very similar and may be merged into one.
R: done
Introduction.
Line 26
"... from the donor at rate ..."
Correction. "... from the donor at a rate ..."
R: done
Lines 37-39
"...however in this article we follow the traditional approach, grouping together those systems with similar light curves but also showing long cycles."
Correction. Start the following as a new sentence.
However in this article we follow the traditional approach, grouping together the systems with similar light curves that also show long cycles."
R: done
Lines 44-45
"... and other of ∼ 270 d was found in UU Cas [14]."
Correction. "... and that of ∼ 270 d was found in UU Cas [14]."
R: done
Lines 45-47
"Guinan [15] inferred a 275 day period for β Lyr attributing a pulsational origin in the B8 II star although he also mention possible changes in the structure of the disk."
Correction. "Guinan [15] inferred a 275 day period for β Lyr attributing a pulsational origin to the B8 II star, although he also mentioned possible changes in the structure of the disk."
R: done
Line 49-50
"... the similarity of β Lyrae with V 1343 (SS 433), a massive X-ray binary of orbital period 13.08 days and bipolar jets."
Correction. "... similarity of β Lyrae with V 1343 Aql (SS 433), a massive X-ray binary with an orbital period of 13.08 days and bipolar jets."
R: done
Line 57
"... a relative massive disk around it ..."
Correction. "... a relatively massive disk around it ..."
R: done
"The gainer is expected to be accelerated rapidly until critical rotation, which should favor the stop of accretion onto the stellar surface and the cumulation of mass in the disk, ..."
Correction. "The gainer is expected to be accelerated rapidly toward critical rotation, which should favor the stop of accretion onto the stellar surface and the accumulation of mass in the disk, ..."
R: done
Section 2.
Line 93
"The long cycle has larger amplitude ..."
Correction. "The long cycle has a larger amplitude ..."
R: done
Lines 110-111
"... but scarce investigations oriented to test the circumstellar matter suggests ..."
Correction. "... but scarce investigations oriented to test the circumstellar matter suggest ..."
R: done
Line 114-116
"The stream of mass flowing from the donor star revolves around the primary, if this is small enough, forming an accretion ring that spreads because of viscosity to rapidly form an accretion disk"
Correction. "The stream of material from the donor star revolves around the primary. If the stream is small enough, it forms an accretion ring that spreads because of viscosity to rapidly form an accretion disk"
R: done
Line 120-121
"... eventually transferring it a large amount of angular momentum and spinning it up" 3
Correction. "... eventually transferring a large amount of angular momentum to it and spinning it up."
R: done
"Many of the disk radius were derived from light curve models susceptibles ..."
Correction. "Many of the disk radii were derived from light curve models susceptible ..."
R: done
Line 132
"... indicate that the disks are thick, more than expected from ..."
Correction. "... indicate that the disks are thicker than expected from ..."
R: done
Line 134
"... motions able to rapidly transport the thermal energy ..."
Correction. "... motions capable of rapidly transporting the thermal energy ..."
R: done
"In this context is valid to ask about the nature of hot and bright zones detected in several systems, through the technique ..."
Correction. "In this context, it is valid to wonder about the nature of hot and bright zones detected in several systems through the technique ..."
R: done
Line 153-154
" ... based on the persistence of blue emission components ..."
Correction. " ... based on persistence of the blue emission components ..."
R: done
Section 4.
Line 164
"The data sampling provided by the ..."
Correction (to avoid use the word "provided" twice in this sentence). "The data sampling from the ..."
R: done
Line 166
"... high-quality dataset useful to explore time variability in time scales ..."
Correction. " ... high-quality dataset useful for exploring temporal variability on time scales ..."
R: done
Line 168
" ... systems have been recently studied revealing interesting clues ..."
Correction. " ... systems have been recently studied and revealed interesting clues ..."
R: done
Line 172-174
" ... assuming fixed some others parameters, usually the mass ratio and donor temperature obtained from the study of spectroscopic data ..."
Correction. " ... assuming some fixed parameters, usually the mass ratio and the donor temperature derived from spectroscopic data ..."
R: done
Line 174-175
" ... or determining the mass ratio with the q photometric method in absence ..."
Question. What is the "q photometric method"? Perhaps, the author meant the following:
" ... or determining the mass ratio q using a photometric method in the absence ..."
R: done
Line 178-179
" ... of not including explicitly stellar or disk winds, neither equatorial mass outflows"
Correction. " ... of neither explicitly including stellar or disk winds nor equatorial mass outflows"
R: done
Line 180-181
" ... by a long cycle length ∼ 307 d and amplitude about 0.8 mag at the I-band."
Correction. " ... by a long cycle length of ∼ 307 d and an amplitude of ~ 0.8 mag in the I-band."
R: done
Line 181-182
" ... The orbital period is 7.75 d and has relatively small stellar masses ..."
Correction. " ... The system has an orbital period of 7.75 d and relatively small stellar masses ..."
R: done
Line 187-188
"This can be understood in terms of changes in temperature and size of the disk, along with changes in their hot and bright spots."
Correction. "This can be understood in terms of changes in the temperature and size of the disk along with changes in the hot and bright spots."
R: done
Line 188-189
"The disk radius at minimum is 189 7.5 R⊙ and at the ascending branch 15.3 R⊙ ..."
Correction. "The disk radius is 7.5 R⊙ at the minimum and 15.3 R⊙ at the ascending branch."
R: done
Line 191
"OGLE-LMC-DPV-065 is other interesting DPV ..."
Correction. "OGLE-LMC-DPV-065 is another interesting DPV ..."
R: done
Line 192-193
"It has similar stellar masses that β Lyrae and orbital period 10.03 d compared with 12.95 d in this last system."
Correction. "It has stellar masses similar to those of β Lyrae and the 10.03 d orbital period compared to 12.95 d in the latter system.
R: done
Line 194-195
" ... at I and V bands, of amplitude about 0.3 and 0.2 mag, respectively, whose general shape is more or less constant, with only minor variability." 5
Correction. " ... in the I and V bands with the amplitudes of ~0.3 and ~0.2 mag, respectively, whose general shape is nearly constant with only minor variations."
R: done
Line 194-195
" ... at I and V bands, of amplitude about 0.3 and 0.2 mag, respectively, whose general shape is more or less constant, with only minor variability."
Correction. " ... in the I and V bands with the amplitudes of ~0.3 and ~0.2 mag, respectively, whose general shape is nearly constant with only minor variations."
R: done
Line 196-197
" ... during about 13 yr, from 350 to 218 d, the long cycle remained almost constant by about 10 yr."
Correction. " ... from 350 to 218 d during about 13 yrs, the long cycle remained almost constant for about 10 yrs."
R: done
Line 198-199
" ... no clear link between disk structure and long cycle is available for this system, in clear contrast ..."
Correction. " ... no clear link between the disk structure and the long cycle has been observed for this system, in a clear contrast ..."
R: done
Line 202-203
"When we look the photometric behaviour of OGLE-BLG-ECL-157529, the situation turns to be again very interesting."
Correction. "When we look at the photometric behaviour of OGLE-BLG-ECL-157529, the situation again turns out to be very interesting."
R: done
Line 208
"... by gainer and the disk."
Correction. "... by the gainer and the disk."
R: done
Line 212-213
" ... longer cycle, becoming smaller than the tidal radius when the long cycle is faster and its amplitude smaller on later epochs ..."
Question. Something is wrong with this sentence. It is unclear how can "longer cycle" become "smaller than the tidal radius".
R: fixed
Line 216-218
"The oscillations of the disk radius in time scales of hundreds of days, cannot be explained by release of viscous energy as occurs with superhumps in precessing disks in cataclysmic variable stars of the SU UMa type, this because Lindblad resonance regions ..."
Correction. ""The oscillations of the disk radius on time scales of hundreds of days cannot be explained by the release of the viscous energy as occurs with superhumps in precessing disks of cataclysmic variable stars of the SU UMa type. This is because Lindblad resonance regions ..."
R: done
Figure 4 caption.
" ... Disentangled long-cycle (up) and orbital (down) ..." --> " ... Disentangled long-cycle (top) and orbital (bottom) ..."
R: fixed
Figure 6 caption. 6
"The tidal radius is indicated by an horizontal dashed line." --> "The tidal radius is indicated by a horizontal dashed line."
R: done
Section 5.
Line 227-229
"The disks are feed by Roche lobe overflow from the donor star and disk changes can in principle be explained as variations in mass transfer rate."
Correction. "The disks are fed by a Roche lobe overflow from the donor star, and the disk changes can in principle be explained by variations in the mass transfer rate."
R: done
Line 229-230
"Zones of low density are also reported, forming Balmer emission lines and revealed in Doppler maps."
Correction. "Zones of low density, where Balmer emission lines form, have also been reported and revealed in Doppler maps."
R: done
Line 232
"Accretion plays a minor role a a source ..."
Correction. "Accretion plays a minor role as a source ..."
R: done
Line 233-234
" ... compared with compact objects where disks are found whose luminosity is accretion driven."
Correction. " ... compared with compact objects, whose disks luminosity is accretion driven."
R: done
Line 242-243
"However, not like β Lyrae, most of these binaries show a constant orbital period and this indicates that β Lyrae is rather unusual object in the class."
Correction. "However, unlike β Lyrae, most of these binaries show a constant orbital period that indicates that β Lyrae is a rather unusual object in the class."
R: done
TRANSLATE with x English
| Arabic | Hebrew | Polish |
| Bulgarian | Hindi | Portuguese |
| Catalan | Hmong Daw | Romanian |
| Chinese Simplified | Hungarian | Russian |
| Chinese Traditional | Indonesian | Slovak |
| Czech | Italian | Slovenian |
| Danish | Japanese | Spanish |
| Dutch | Klingon | Swedish |
| English | Korean | Thai |
| Estonian | Latvian | Turkish |
| Finnish | Lithuanian | Ukrainian |
| French | Malay | Urdu |
| German | Maltese | Vietnamese |
| Greek | Norwegian | Welsh |
| Haitian Creole | Persian |
Reviewer 2 Report
Report on #1511475:
"Accretion Disks and Long Cycles in beta-Lyrae type Binaries"
by Ronald E. Mennickent
This is an excellent review paper about the observational
properties of the light curves of beta Lyrae binaries,
systems observed during a key stage of mass transfer.
These systems often display photometric variations on
both the orbital and longer timescales, and the author,
Dr. Mennickent, has been a pioneer in observational studies
of these systems. I recommend the paper for publication
in the special issue "What’s New under the Binary Suns."
In the attached PDF file, I have marked a number of places
that merit revision before publication.
If page limits are not prohibitive, then Dr. Mennickent
might consider adding some text about precession models
for the long periods. Disk precession is a compelling
explanation for the long periods because it causes a
periodic variation in projected surface area associated
with the flux variations. There has been much work on disk
dynamics over the last decade that may be relevant for
these binaries. For example, Dogan et al. (2015, MNRAS,
449, 1251) discuss disk precession and implications for
accretion. Larwood et al. (1996, MNRAS, 282, 597)
discuss models of disk warping and precession. Their
equation 22 gives the predicted precession period for
a disk with a constant surface density and given disk
radius. This can be re-expressed as the ratio of the
precession period to the orbital period:
P_{prec} / P_{orb} = {32\over 15} (q(q+1)/r_d^3)^{1/2}
where
q = Mass(gainer) / Mass(donor)
r_d = disk radius / semimajor axis
= f R_{Roche}
f = disk fill out factor
R_{Roche) = fractional Roche radius of gainer star.
For example, most of the systems in Table 1 have q = 4.
Using the Roche radius estimator from Eggleton (1983,
ApJ, 268, 368) and q = 4, I find:
f (%) P_{prec} / P_{orb}
80 38
90 31
100 27
These ratios are comparable to the P_l / P_o ratios shown
in the histogram of Figure 2 (right panel). Thus, the
long periods appear to be consistent with expectations
for precessing disks. The precession model predicts that
the long periods will be larger with higher mass ratio
q and with smaller disk radius r_d, and it would be
interesting to see if these trends are found in the
sample of known Double Periodic Variables.

Author Response
We thank the referee for the time invested in reviewing the first version of this manuscript and for the useful comments on it.
Specially we thank the comments on disk precession and the corrections in the PDF file. We have added a paragraph at the end of section 2 and some lines at the end of section 5 dealing with the issue of precessing disks. We have also incorporated all the corrections in bold face in the new version as detailed in the list below:
i.e., [here and elsewhere add comma]
R: commas were added
has reverted > has reversed
R: done
mass ratio as a result of the
R: done
delete “been” ; delete “in”
R: done
as a prototype of a single
R: done
the accumulation
R: done
suggests a promising role of this
R: done
Many measurements of the
R: done
susceptible
R: done
methods sensitive to
R: done
technique sensitive to
R: done
is related to
R: done
continuous
R: done
with a disk was used to find the
R: done
assuming some fixed parameters, usually
R: done
winds, nor equatorial
R: done
given as 306 days in Table 1
R: fixed
and the system has relatively small
R: done
the system is brighter in the first quadrant than during the second quadrant.
R: done
is another interesting
R: done
and the long
R: done
radius on timescales of
R: done
days cannot [remove comma]
R: done
type, because the Lindblad
R: done
shown as gray points for comparison.
R: done
long cycle (bottom) and orbital (top) light
R: done
zones, and in a few cases
R: done
donor star, and disk changes
[add comma]
R: done
role as a source of disk luminosity
R: done
However, unlike $\beta$
R: done
period, and
[add comma]
R: done
is a rather unusual
R: done
obvious similarities -
R: this part of the text was modified in response to another referee
hot Algols showing
R: this part of the text was modified in response to another referee
all systems, and the
[add comma]
R: done
the long cycle deserves further investigation.
R: done
binary beta Lyr A
R: done
I think the legend is reversed: stars represent the larger edge thickness.
R: figure is fine, disk central thickness is larger than disk edge thickness in this object
TRANSLATE with x English
| Arabic | Hebrew | Polish |
| Bulgarian | Hindi | Portuguese |
| Catalan | Hmong Daw | Romanian |
| Chinese Simplified | Hungarian | Russian |
| Chinese Traditional | Indonesian | Slovak |
| Czech | Italian | Slovenian |
| Danish | Japanese | Spanish |
| Dutch | Klingon | Swedish |
| English | Korean | Thai |
| Estonian | Latvian | Turkish |
| Finnish | Lithuanian | Ukrainian |
| French | Malay | Urdu |
| German | Maltese | Vietnamese |
| Greek | Norwegian | Welsh |
| Haitian Creole | Persian |
Reviewer 3 Report
Referee report on the review paper
ACCRETION DISKS AND LONG CYCLES IN BETA-LYRAE TYPE BINARIES
by R. E. Mennickent
This is a very nice and conscise review about our present knowledge about the massive, double periodic Algol-like eclipsing binaries known as beta Lyr (EB) type systems. I have no any fundamental objections or concerns about the scientific content of the present paper. Thus, in my view, the manuscript is almost ready for publications, I found only a few issues that should be corrected, as follows:
Line 49: The correct GCVS designation of SS433 is V1343 Aquilae (the nem of the constellation is missing in the submitted manuscript, however, I am certain that it is only a simple typo).
Figure 3: In the caption "black points" are mentioned, however, I do not see black points in the plot. I guess, the Author intended to say "gray points".
Figure 4: The caption contradicts the arrangement of the upper and lower panels. Either the caption or the order of the panels should be modified accordingly.
Finally, about the question about the appropriateness that beta Lyr should be considered as a prototype of a class of eclipsing binaries or, not, I have a minor comment. This question is mentioned by the Author both in the Introduction (lines 31-39), and in the Conclusions (lines 239-250). At both places he refer to the recent review paper of Wilson, 2020, Galaxies, 8, 57. In my view, referencing this paper is appropriate in the context of the Introduction, but not of the Conclusions. The reason is, that Wilson says that there are no any astrophysical reasons for separate beta Lyraes from classic Algols, i.e. there is no need for the class of beta Lyr-s. It is the same context that is mentioned in the Introduction (i.e., that from an evolutionary point of view, beta Lyr-s are Algol systems). In contrast to this in the Conclusions it was mentioned that beta Lyr itself physically differ significantly from most of the other systems categorized as beta Lyr-s and, therefore, beta Lyr itself is far from being an appropriate prototype of beta Lyr-type systems. While this latter statement is naturally correct, in my view it is not the same context that was applied in Wilson's paper.
Author Response
We thank the referee for the time invested in reviewing and for the useful comments on the first version of this manuscript. Here the reply to the points mentioned by the referee.
Line 49: The correct GCVS designation of SS433 is V1343 Aquilae (the neme of the constellation is missing in the submitted manuscript, however, I am certain that it is only a simple typo).
R: we added “Aquilae”
Figure 3: In the caption "black points" are mentioned, however, I do not see black points in the plot. I guess, the Author intended to say "gray points".
R: we changed “black” by “gray”
Figure 4: The caption contradicts the arrangement of the upper and lower panels. Either the caption or the order of the panels should be modified accordingly.
R: We modified the caption accordingly
Finally, about the question about the appropriateness that beta Lyr should be considered as a prototype of a class of eclipsing binaries or, not, I have a minor comment. This question is mentioned by the Author both in the Introduction (lines 31-39), and in the Conclusions (lines 239-250). At both places he refer to the recent review paper of Wilson, 2020, Galaxies, 8, 57. In my view, referencing this paper is appropriate in the context of the Introduction, but not of the Conclusions. The reason is, that Wilson says that there are no any astrophysical reasons for separate beta Lyraes from classic Algols, i.e. there is no need for the class of beta Lyr-s. It is the same context that is mentioned in the Introduction (i.e., that from an evolutionary point of view, beta Lyr-s are Algol systems). In contrast to this in the Conclusions it was mentioned that beta Lyr itself physically differ significantly from most of the other systems categorized as beta Lyr-s and, therefore, beta Lyr itself is far from being an appropriate prototype of beta Lyr-type systems. While this latter statement is naturally correct, in my view it is not the same context that was applied in Wilson's paper.
R: following referee’s comment we exclude the mentioned reference in the Conclusions. We changed the text:
“An alternative explanation is that $\beta$ Lyrae, in spite of the obvious similitudes - disk around a B-type gainer, stage of mass transfer from a cooler and less massive star - is not a prototype of the group of hot algols showing long photometric cycles reviewed in this article. The inappropriateness of using $\beta$ Lyrae as a prototype of binaries with EB type light curves has been risen by \citet{2020Galax...8...57W} and could be relevant here.”
by
“Therefore $\beta$ Lyrae itself is far from being an appropriate prototype of $\beta$-Lyrae type systems.”
TRANSLATE with x English| Arabic | Hebrew | Polish |
| Bulgarian | Hindi | Portuguese |
| Catalan | Hmong Daw | Romanian |
| Chinese Simplified | Hungarian | Russian |
| Chinese Traditional | Indonesian | Slovak |
| Czech | Italian | Slovenian |
| Danish | Japanese | Spanish |
| Dutch | Klingon | Swedish |
| English | Korean | Thai |
| Estonian | Latvian | Turkish |
| Finnish | Lithuanian | Ukrainian |
| French | Malay | Urdu |
| German | Maltese | Vietnamese |
| Greek | Norwegian | Welsh |
| Haitian Creole | Persian |